# Immunoproteasome Function in Normal and Malignant Hematopoiesis

**DOI:** 10.3390/cells10071577

**Published:** 2021-06-22

**Authors:** Nuria Tubío-Santamaría, Frédéric Ebstein, Florian H. Heidel, Elke Krüger

**Affiliations:** 1Innere Medicine C, Universitätsmedizin Greifswald, 17475 Greifswald, Germany; nuria.tubiosantamaria@med.uni-greifswald.de; 2Institut für Biochemie und Molekularbiologie, Universitätsmedizin Greifswald, 17475 Greifswald, Germany; ebsteinf@uni-greifswald.de; 3Leibniz Institute on Aging, Fritz Lipmann-Institute, 07745 Jena, Germany

**Keywords:** ubiquitin–proteasome system (UPS), immunoproteasome (iP), proteasome inhibitors (PIs), hematopoiesis, hematologic malignancies

## Abstract

The ubiquitin–proteasome system (UPS) is a central part of protein homeostasis, degrading not only misfolded or oxidized proteins but also proteins with essential functions. The fact that a healthy hematopoietic system relies on the regulation of protein homeostasis and that alterations in the UPS can lead to malignant transformation makes the UPS an attractive therapeutic target for the treatment of hematologic malignancies. Herein, inhibitors of the proteasome, the last and most important component of the UPS enzymatic cascade, have been approved for the treatment of these malignancies. However, their use has been associated with side effects, drug resistance, and relapse. Inhibitors of the immunoproteasome, a proteasomal variant constitutively expressed in the cells of hematopoietic origin, could potentially overcome the encountered problems of non-selective proteasome inhibition. Immunoproteasome inhibitors have demonstrated their efficacy and safety against inflammatory and autoimmune diseases, even though their development for the treatment of hematologic malignancies is still in the early phases. Various immunoproteasome inhibitors have shown promising preliminary results in pre-clinical studies, and one inhibitor is currently being investigated in clinical trials for the treatment of multiple myeloma. Here, we will review data on immunoproteasome function and inhibition in hematopoietic cells and hematologic cancers.

## 1. Introduction

The ubiquitin–proteasome system (UPS) is the main non-lysosomal pathway for the degradation of intracellular proteins. It consists of a sequence of enzymatic processes that tag a protein substrate with multiple ubiquitin molecules for subsequent degradation by the 26S proteasome, with the release of reusable ubiquitin performed by deubiquitinating enzymes (DUBs) (Figure 1) [1]. During the initial step of ubiquitin conjugation, a ubiquitin-activating enzyme (E1) activates ubiquitin in an ATP-dependent manner. One of the several ubiquitin-conjugating enzymes (E2) transfers the activated ubiquitin to the substrate, which is specifically bound to a ubiquitin-ligase enzyme (E3), in a two-step reaction [2]. Hundreds of E3 enzymes have been characterized by individually defining motifs, determining high substrate specificity [3]. Monoubiquitylation, namely, the attachment of only one ubiquitin moiety through its C-terminal carboxylate to a protein, can regulate endocytosis, endosomal sorting, histone regulation, and DNA repair [4]. Importantly, ubiquitin itself exhibits eight potential sites for ubiquitination (M1, K6, K11, K27, K29, K33, K48, and K63), thereby allowing the formation of polyubiquitin chains. Polyubiquitination through different ubiquitination linkages may carry distinct biological fates, with proteins modified with K48- or K11-linked polyubiquitin being typically degraded by the 26S proteasome [4]. The 26S proteasome is a 2.5 MDa complex that consists of a 20S core particle and a 19S regulatory particle [5,6,7]. It functions as a crucial regulator of the proteome in eukaryotic cells by degrading damaged, misfolded, or regulatory proteins [8,9].

Besides the constitutive 26S proteasome, which is expressed in all different tissues and cell types, three other proteasome isoforms, the immunoproteasome (iP), the thymoproteasome, and the spermatoproteasome, are expressed in a tissue-dependent manner [10,11]. In the case of the iP, the three main catalytic subunits of the constitutive proteasome (β1, β2, β5) are substituted by the so-called immunosubunits (β1i, β2i, β5i) [12,13,14]. The iP subunits appeared during the two rounds of whole-genome duplication that occurred before the emergence of the common ancestor of jawed vertebrates [11]. The iP is constitutively expressed in cells of hematopoietic origin or can be induced after cytokine stimulation in various tissues [15]. It modulates MHC class I antigen processing [16,17] and supports the differentiation of T-helper cells in the context of virus infection [18]. A later discovered catalytic subunit (β5t) was described to be exclusively expressed in cortical thymic epithelial cells, suggesting that the thymoproteasome has a key role in generating the MHC class I antigen repertoire during thymic selection [19]. The β5t subunit emerged from the β5 in a common ancestor of jawed vertebrates [11]. Finally, the most dramatic changes in tissue-specific proteasome composition have been observed in the testis. In mammalian cells, a testis-specific variant in the α4s subunit of the spermatoproteasome [20] has been found to be essential for the assembly of the proteasome regulator PA200. PA200 is a different type of regulatory particle that can bind to the 20S particle instead of the 19S cap, and it is particularly abundant in testes [21]. It has been shown to promote the acetylation-dependent degradation of core histones during somatic DNA damage responses and spermatogenesis [22,23].

Protein homeostasis relies on protein degradation by the UPS, which is necessary to maintain the functionality of the hematopoietic system [24,25]. Hematopoietic stem cells (HSCs) give rise to all mature blood cells through a hierarchical process called hematopoiesis [26]. This process requires tight regulation of quiescence, self-renewal, and differentiation [27,28]. Dormant HSCs show low protein synthesis [29], which increases under stress conditions and may lead to the production of misfolded or denatured proteins [30]. To prevent their aggregation, which would cause harmful effects, these damaged proteins must be efficiently degraded by the UPS. Here, alterations of ubiquitin-dependent proteolysis of cell-cycle regulators or house-keeping genes have been shown to result in malignant transformation [31,32,33,34]. The growing recognition of fundamental UPS functions has prompted the search for pharmacologic inhibitors to inactivate this pathway, making it an attractive therapeutic target [35,36].

Proteasome inhibitors (PIs) have been approved by the US Food and Drug Administration for the treatment of patients with hematopoietic cancers, especially for multiple myeloma (MM) [37,38]. However, the acquisition of resistance and toxicity (including pain, fatigue, peripheral neuropathy, myelosuppression, and cardiotoxicity) remain a clinical challenge [39]. Recently, next-generation proteasome inhibitors that may overcome resistance to first-generation compounds [40,41,42] have been developed, but these also show adverse effects on normal cells.

In general, proteasome inhibition can target any of the three proteasome proteolytic sites—the caspase-like (β1), trypsin-like (β2), and chymotrypsin-like (β5) sites—whereby most of the PIs developed so far inhibit the β5/β5i subunits of the constitutive proteasome (cP) and the iP. It remains unclear whether the specific thymoproteasome subunit β5t is also affected by established PIs; its sensitivity to standard PIs has been shown to substantially differ from those of β5/β5i [43]. Differences in pharmacokinetics due to different chemical backbones explain the diversity of PIs with regard to their activity, safety, and tissue distribution [35]. Furthermore, selective inhibitors of the iP subunits that target both the cP and the iP have been developed as an alternative to PIs [44,45]. This review will focus on the role of the iP in the hematopoietic system and in malignant transformation and compile information on iP inhibitors currently investigated in pre-clinical or clinical studies for the treatment of hematologic malignancies.

## 2. The Immunoproteasome: A Proteasomal Variant Linked to the Hematopoietic System

### 2.1. Immunoproteasome Structure

The 26S constitutive proteasome structurally consists of a catalytic 20S core particle with three different peptidase activities and one or two terminal 19S regulatory particles composed of six ATPases and multiple components necessary for substrate binding (Figure 2a) [5,6,7]. The 19S particle binds to one or both ends of the 20S proteasome and, together, they form the enzymatically active 26S proteasome.

The composition of the 20S particle consists of four rings with seven subunits each. The two outer rings contain α-subunits and the inner ones β-subunits. Three of the seven β-subunits, the β1, β2, and β5 subunits, are responsible for the enzymatic activities of the proteasome with caspase-like, trypsin-like and chymotrypsin-like activities, respectively (Figure 2a). The 19S particle, also known as PA700, serves as a gate to the 20S particle and can be divided into base and lid subcomplexes [46]. The base contains six ATPases (Rpt1-6) that unfold substrates prior to translocation into the 20S core particle, while the lid is required for recognition of the ubiquitin-modified proteins, notably via the Rpn10 [47], Rpn13 [48], and Rpn1 [49] subunits.

The iP contains the three de novo synthesized subunits, β1i (encoded by the *PSMB9* gene in humans and by the *LMP2* in mice), β2i (encoded by *PSMΒ10*/*MECL1*), and β5i (encoded by *PSMB8*/*LMP7*), that substitute the constitutive ones (Figure 2a). The β1i/*LMP2* and β5i/*LMP7* genes are located in the MHC class II genomic region [13,14], which led to the term “immunosubunits”. After INF-γ stimulation, β1i and β5i subunits are expressed and incorporated into proteasome precursor complexes instead of their homologous counterparts β1 and β5 [50,51]. Later, the third INF-γ inducible protein was identified as β2i/*MECL1* and also found to replace the standard β2 subunit [12]. It has been shown that direct interaction of β5i with the assembly chaperone proteasome maturation protein (POMP) accelerates iP biogenesis to the detriment of cP assembly, allowing a quick response to immune and inflammatory stimuli [52]. Surprisingly, the simultaneous expression of constitutive and inducible β-subunits is possible, thereby setting a variety of different 20S complexes from which the β1/β2/β5i and β1i/β2/β5i combinations are the most common ones [53]. These proteasome variants are collectively known as intermediate or mixed-type proteasomes [54].

Moreover, INF-γ stimulation induces the expression of the proteasome activator 28 (PA28), a heteroheptameric complex that binds to the cP or the iP with the same affinity [55]. PA28 can associate with one or both ends of a 20S particle or to the free end of a 19S–20S complex to form homo (PA28-20S) or hybrid (PA28-20S–19S) complexes, respectively [56]. While it is understood that PA28-20S proteasome complexes primarily degrade oxidant-damaged proteins in a ubiquitin-independent manner [57,58], hybrid proteasomes seem to specialize in the supply of MHC class-I-restricted peptides [59,60,61].

### 2.2. Immune and Non-Immune Functions of the Immunoproteasome

As discussed above, the proteasome functions as a key modulator and central part of the UPS. Of note, it plays a dual role by exerting immune and non-immune functions. It not only generates antigen peptides for immune responses but also degrades damaged or misfolded proteins produced under stress stimuli and short-lived proteins with regulatory functions in cell differentiation, cell-cycle regulation, transcriptional regulation, or apoptosis [8,9] and thereby facilitates regulation of intrinsic cell processes.

The proteasome is the main protease involved in the generation of antigenic peptides presented by MHC class I molecules to cytotoxic T-lymphocytes [62]. Expression of the immunosubunits (β1i, β2i, and β5i) after INF-γ induction has been shown to modulate the efficiency of peptide production, generating peptides that are better suited to bind MHC class I molecules, [16,17] by exhibiting a different cleavage rate [63]. Herein, depending on protein sequence, some antigens are exclusively produced by the iP or the cP, while others can be processed by both [64,65,66,67,68]. There are even some antigens that seem to be preferentially processed by intermediate-type proteasomes [53,69]. Moreover, mice deficient in β1i/LMP2, β5i/LMP7, or β2i/MECL1 exhibit modest defects in MHC class I antigen presentation [70,71,72], while mice deficient for all iP subunits have an impairment in the presentation of MHC class I epitopes, similar, regarding the immunological phenotype, to β5i/LMP7-deficient mice [73]. The iP has additional immunological functions apart from MHC class I antigen processing. First, the iP plays a role in the maintenance and expansion of the CD8+ T-cell repertoire during immune response against intracellular infections [18,74]. Additionally, it promotes the differentiation of pro-inflammatory T-helper type 1 (Th1) and type 17 (Th17) cells and suppresses the induction of regulatory cells [75,76,77]. It also induces the production of cytokine IL-23 by monocytes and IL-2 by T-cells [45].

In contrast to the initial assumption that the iP only plays a specific role in MHC class I antigen production, it is now accepted that this function is part of a more general role in protein homeostasis. The iP has been shown to prevent the accumulation of harmful protein aggregates under cytokine-induced oxidative stress due to increased efficiency in protein degradation compared to the cP [78,79,80,81,82], even though this aspect is still a matter of debate [83,84]. In addition, the 20S core particle has been demonstrated to dissociate from the 26S proteasome under stress conditions, with the iP containing 20S being more efficient than its standard counterpart at degrading oxidized proteins in an ATP- and ubiquitin-independent manner [58,85]. Supporting this notion, expression of the iP is upregulated through the mTOR pathway to prevent the accumulation of misfolded or damaged proteins [82].

The fact that iPs are more efficient than cPs at breaking down intracellular proteins implies that these may exert pleiotropic effects on cell function. As the proteasome has been associated with control of transcription [86], the iP may also impact transcription during cell stress or malignant transformation. Along these lines, iP expression has been shown to modulate the abundance of transcription factors that regulate fundamental signaling pathways [87,88].

### 2.3. Expression Patterns of the Immunoproteasome

While cP is highly expressed in various tissues and its constitutive expression and formation is controlled on the transcriptional level through Nrf1/Tcf11 or inducible mainly upon proteotoxic stress, iP subunits are downregulated under these conditions [89,90,91]. Expression of the three iP subunits is inducible after pro-inflammatory cytokine stimulation in many tissues. IP induction is mediated by the activation of signal transducer activator of transcription (STAT) and interferon regulatory factor (IRF) families [92,93,94]. Moreover, iP subunits are constitutively expressed at high levels in cells of hematopoietic origin across different species [95,96] and can be found in vivo in hematopoietic cells such as macrophages [97] or B-cells [98]. The iP is the predominant proteasome variant found in the bone marrow cells of healthy individuals and MM patients [41]. Similarly, in tumor cell lines of hematopoietic origin, the iP represents the major constituent of the total proteasome pool, while in other non-hematologic-derived cells, the percentage of iPs appeared rather low [41]. Of note, the ratio of iP-to-cP expression is significantly higher in pre-B acute lymphoblastic leukemia (ALL) than in acute myeloid leukemia (AML) pediatric patients. This ratio correlates with therapy response to PIs, suggesting it can be used as an indicator of sensitivity [99,100]. Immunoproteasome expression has also been found to be upregulated in other types of hematologic malignancies, such as in myeloproliferative neoplasms (MPNs), specifically in primary myelofibrosis (PMF) [101]. Apart from its expression in hematopoietic cells, the β5i/*LMP7* subunit is also expressed in small intestinal epithelial cells [102], colon [103], liver [104,105], umbilical vein cells [106], and placenta [107].

The β5i/*LMP7* subunit is known to be incorporated into 20S proteasome assembly intermediates, preferentially by higher affinity to the assembly factor POMP [52]. In some cell types, the exclusive expression of β5i/*LMP7* leads to the formation of intermediate type proteasomes, with only one or two immunosubunits incorporated [53]. Such proteasome subtypes were shown to generate spliced tumor epitopes more efficiently than other subtypes [108]. POMP expression, in turn, is negatively controlled by micro-RNA miR-101 to modulate (immuno-)proteasome formation. Manipulation of miR-101 is engaged by breast cancer cells to ensure higher proteasome activity along with higher proliferation rates [109]. miRNAs—including miR-101 and others—are associated with cancer immunity [110] and thus discussed as a potential therapeutic target. In this context, miR-101 is also proposed to target Jak2 [111].

### 2.4. Genetic Variants of the Immunoproteasome

All proteasome subunits have known genetic variants, with some of them causing diseases. Variants in the *PSMA6* gene have been linked to coronary artery disease, myocardial infarction, type II diabetes mellitus, and ischemic stroke, while an association between variants in the *PSMA7* gene and intellectual disability has been reported [112]. Recently, two reports have associated variants in the *PSMC3* gene with severe congenital deafness, early-onset cataracts, and various neurological features [113] and *PSMΒ1* variants with microcephaly, intellectual disability, developmental delay, and short stature [114]. Polymorphisms in genes encoding the iP subunits β1i (*PSBM9*) and β5i (*PSMB8*) have been associated with an increased risk of tumor development, including the development of esophageal carcinoma [115], cervical carcinoma [116], oral squamous cell carcinoma [81,117], prostate cancer [118], and colon cancer [119]. It has also been reported that polymorphisms in *PSBM9* but not in *PSMB8* can be used as a susceptibility factor in the development of AML or MM [120]. The G201V mutation in the *PSMB8* gene has been reported in Nakajo–Nishimura syndrome [121]. During iP biogenesis, the mutated β5i protein is not correctly incorporated into mature proteasomes, leading to reduced proteasome activity and an accumulation of ubiquitinated proteins within the cells that highly express iPs. Another mutation in *PSMB8* (G197V) has been found in patients with an autoinflammatory disease [122]; it produces a significant decrease in proteasome function and an accumulation of ubiquitin-modified proteins in the patient’s tissues. Meanwhile, further mutations in the *PSMB8* gene or other genes coding for iP or cP subunits, including *PSMB9*, *PSMB10*, *PSMA3, PSMB4, POMP,* and *PSMG2*, have been identified in patients suffering from similar autoinflammatory syndromes [123,124,125,126,127]. Because of their clinical manifestations and proteasomal etiology, these conditions are frequently referred to as chronic atypical neutrophilic dermatosis with lipodystrophy and elevated temperature (CANDLE) or proteasome-associated autoinflammatory syndrome (PRAAS), respectively. On the molecular level, cells suffer from severe proteotoxic stress in these conditions, including ER-stress or activation of Nrf1/Tcf11 [89,90,126], similar to the cells treated with proteasome inhibitors. 

## 3. Pro- and Anti-Tumoral Properties of the Immunoproteasome

The iP has been found to exert antagonistic effects on different types of malignancies, having pro- or anti-tumorigenic properties that are all due to its capacity to modulate the expression of pro-tumorigenic cytokines and chemokines or to increase the presentation of tumor peptides, respectively.

The loss of MHC class I molecules on the tumor surface is a well-known mechanism for evading recognition and destruction by cytotoxic T-lymphocytes [128,129]. An alternative strategy for escaping immune control has been found in non-small cell lung cancer (NSCLC), in which epithelial-to-mesenchymal transition leads to a loss of iP expression, resulting in markedly reduced MHC class I antigen presentation [130] and cancer progression. Conversely, higher iP expression correlates with improved prognosis. Here, increased iP expression was attributed to the secretion of INF-γ by CD8+ tumor-infiltrating lymphocytes (TILs), the presence of which is considered a good prognosis factor [131,132,133]. A similar mechanism has also been reported in melanoma, where high β5i/*PSMB8* and β1i/*PSMB9* expression has been associated with increased survival, enhanced immune response, and the presence of TILs [134]. In breast cancer, high expression of iP subunits correlates with good prognoses and an abundance of TILs [135,136]. On the other hand, iP expression is essential for the initiation of inflammatory processes [137], which can lead to inflammation-driven carcinogenesis. The iP seems to play a fundamental role in colitis-associated carcinogenesis (CAC) since increased expression of β5i/*LMP7* and β1i/*LMP2* has been observed in inflamed colons and *LMP7*-deficient mice exhibit a reduction in tumor formation compared to their wild-type counterparts [138]. *LMP7* deficiency also leads to reduced expression of pro-tumorigenic chemokines CXCL1, CXCL2, and CXCL3 and decreased secretion of IL-6 and TNF-α in this model.

Independently from its pro- or anti-tumorigenic properties, iP expression in these types of cancer is dependent on the paracrine production of pro-inflammatory cytokines by the surrounding immune cells, as particularly exemplified by breast cancer cells [136]. On the contrary, as mentioned above, constitutive expression of iPs without stimulation has mainly been found at high levels in cells of hematopoietic origin, including hematologic malignancies. This high iP expression may indicate a dependency of hematologic malignant cells to iP function, which one begins to explore with the development of selective iP inhibitors. However, the sensitivity of different types of hematologic malignancies may vary depending on iP expression or may be oncogene-specific. For instance, it was found that acute promyelocytic leukemia (APL), which contains the chromosomal translocation PML/RARα, can evade immune control by suppressing the PU.1-dependent activation of immunosubunits [139].

## 4. Development of Immunoproteasome Inhibitors to Target Hematologic Malignancies

The first PIs developed were non-selective and inhibited both cPs and iPs to the same extent. Building on these rather unspecific compounds, chemical modifications have led to the alteration of their structure and selective binding capacity to specific proteasome subunits (Figure 2b).

Non-selective inhibitors of the proteasome have proved their efficacy against MM and other hematologic malignancies. Compounds such as bortezomib, carfilzomib, and ixazomib have already been approved for clinical use and later-generation inhibitors are currently being investigated in advanced clinical trials [140,141]. The fact that these inhibitors indiscriminately target both the cP and the iP may lead to a non-selective inhibition of protein degradation. This lack of specificity may account, in part, for the side effects that are often observed, as well as drug-resistance relapse following long-term treatment. Bortezomib-resistant MM cells often show manipulation of proteasome subunits and their expression, although the molecular mechanisms of resistance are not entirely clear. MM cells from patients with relapses present mutations in the β5 subunit, different proteasome subunit compositions, and induction of proteasome subunits. Since all mutations detected after bortezomib treatment are in the binding site of β5 to the inhibitor, the development of iP inhibitors will solve at least the problem with β5 active site mutations [142]. The specificity of binding of PIs to the cP and/or the iP subunits is determined by interactions with the substrate-binding channels (S1, S2, and S3) of the proteolytic subunits [143]. By X-ray crystallography, it was determined that the immunosubunits (β1i, β2i, β5i) present substitutions, in particular, the amino acids of the substrate-binding channels, compared to constitutive subunits (β1, β2, β5). Exploiting these differences allows for the development of specific cP or iP inhibitors [143].

Selective inhibition of the iP over the cP may overcome side effects while maintaining anti-myeloma or anti-lymphoma efficacy. Since the iP is the major proteasome form expressed in cells of hematopoietic origin, including MM cells [41], treatment with iP inhibitors could spare other tissues with little or no iP expression. Moreover, some reports have shown that normal hematopoietic cells may be able to better overcome the effects of iP inhibition. The treatment of mouse splenocytes with an iP inhibitor led to reduced MHC-I surface expression in lymphocytes but did not affect the viability of the cells [45]. In agreement, iP inhibition in naïve T- and B-cells, which express almost exclusively iP or mixed-type proteasomes, leads to mild proteostasis stress, with T-cells being able to recover without increased apoptosis [55], which is also true for microglia [144]. Moreover, normal PBMCs had a minimal reduction of viability after iP inhibitor treatment [145]. In contrast, other studies have pointed out a reduction of viability in PBMCs after iP inhibition [44]. Additionally, because the iP is induced by pro-inflammatory cytokines during stress conditions, iP inhibitors are also an appealing therapeutic target in inflammatory and autoimmune diseases [146,147]. In fact, great progress has been made regarding therapy for these diseases, with iP inhibitors and several novel inhibitors currently being investigated in advanced clinical trials. A list of all the iP inhibitors developed to date can be found in Table 1. In contrast, and despite the encouraging preliminary results, the development of iP inhibitors for the treatment of hematologic malignancies is rather slow-paced. In this section, we present a comprehensive review of iP inhibitors and their anti-tumoral effects in hematologic malignancies (Table 2).

**UK-101** is a dihydroeponemycin analog developed to selectively inactivate the β1i subunit of the iP [148]. Initially, prostate cancer cell lines with a high content of β1i were shown to be more sensitive to UK-101 treatment than cell lines with low β1i expression [148]. More recent results have confirmed that in vivo UK-101 treatment reduces tumor growth in a xenograft model of prostate cancer [149]. Moreover, UK-101 has been demonstrated to reduce cell growth in MM patient samples, even in cells that have become resistant to bortezomib treatment [150].

**ONX-0914 (PR-957)** is the first selective iP inhibitor developed against the β5i subunit, and it has been described as a peptide–ketoepoxide related to the cP inhibitor carfilzomib [45]. Using ONX-0914, several reports have shown the therapeutic potential of iP inhibition in various inflammatory and autoimmune diseases, including rheumatoid arthritis [45], multiple sclerosis [151], colitis [137], and lupus [152]. In vitro ONX-0914 treatment in ALL and AML patient samples have determined that ALL samples are more sensitive to iP inhibition than AML samples (LC_50_ for ALL was 44.6 nM and for AML 248 nM). An increased ratio of immunoproteasome/constitutive proteasome expression was correlated with increased sensitivity to iP inhibitor treatment [99]. Among different AML subtypes, MLL-rearranged (MLLr) AML had the highest iP expression. Treatment of an MLLr cell line with ONX-0914 led to decreased viability and accumulation of polyubiquitinylated proteins, while another AML cell line containing a different chromosomal rearrangement was unaffected by the treatment [136]. Both cell lines were sensitive to non-selective proteasome inhibition with bortezomib and MG132, suggesting that a higher iP expression renders cells more sensitive to iP inhibition. 

Other reports have indicated that MM cell lines exhibit reduced proliferation after ONX-0914 treatment [153]. Increasing the expression of iPs by INF-γ treatment made MM cells more sensitive to ONX-0914 but could not increase sensitivity to the cP inhibitor carfilzomib. Combined treatment of ONX-0914 with a β2 inhibitor dramatically sensitized MM cell lines and primary patient samples to ONX-0914. Similarly, ONX-0914 synergized with cP inhibitors in vitro and in vivo [153].
cells-10-01577-t001_Table 1Table 1Immunoproteasome inhibitors in pre-clinical and clinical development.InhibitorDeveloped byBackboneTargetBinding KineticsUK-101[148]Peptidyl epoxyketoneβ1i subunit (144-fold more selective than β1, but only 10-fold to β5)Covalent irreversibleONX-0914 (PR-957)[45]Peptidyl epoxyketoneβ5i subunit (20- to 40-fold more selective than β5 or β1i)Covalent irreversibleIPSI-001[44]Peptidyl aldehydeβ1i subunit (100-fold more selective than β1)Covalent reversiblePR-924[41]Peptidyl epoxyketoneβ5i subunit (130-fold more selective than β5)Covalent irreversibleLU-001i[154]Peptidyl epoxyketoneβ1i subunit (925-fold more selective than β1)Covalent irreversibleLU-015i[154]Peptidyl epoxyketoneβ5i subunit (553-fold more selective than β5)Covalent irreversibleLU-035i[154]Peptidyl epoxyketoneβ5i subunit (500-fold more selective than β5)Covalent irreversibleHT2210 and HT2106[155]Oxathiazoleβ5i subunit (>4700-fold more selective than β5c)Covalent irreversible1-CA and 4-CA[156]Peptidyl epoxyketoneβ5i subunit (75- to 150-fold more selective than β5)Covalent irreversiblePKS2279 and PKS2252[157]N,C-capped dipeptideβ5i subunit (5600 and 13,600-fold more selective than β5)Non-covalentKZR-504[158]Peptidyl epoxyketoneβ1i subunit (925-fold more selective than β1)Covalent irreversibleKZR-616[158]Peptidyl epoxyketoneβ5i and β1i subunits (18- and 81-fold more selective than β5 and β1c)Covalent irreversibleLU-002i[159]Peptidyl epoxyketoneβ2i subunitCovalent irreversibleM3258[160]Boronic acidβ5i subunit (>500-fold more selective than β5)Covalent reversible
cells-10-01577-t002_Table 2Table 2In vitro and in vivo testing of immunoproteasome inhibitors in hematologic malignancies.InhibitorEffective againstIn Vitro/In Vivo ExperimentsReferencesUK-101MMPatient samples[150]ONX-0914(PR-957)MM and MLLr-AMLHuman cell linesSynergism with BTZ in a MM murine model[99,127,136]IPSI-001MM, NHL, CLL, and AMLHuman cell lines and patient samples[44]PR-924MM, T-ALL, and AMLHuman cell lines and patient samplesMM xenograft model[79,145]LU-035iMMHuman cell lines in conjugation with cytotoxic drug[161]HT2210 and HT2106NHLHuman cell lines[155]M3258MM, AML, and lymphomaHuman cell linesMM xenograft modelPhase I clinical trial[160,162,163]MM: multiple myeloma; MLLr: MLL rearranged; AML: acute myeloid leukemia; NHL: non-Hodgkins lymphoma; CLL: chronic lymphocytic leukemia; T-ALL: T-cell acute lymphoblastic leukemia; BTZ: bortezomib.


**IPSI-001** was selected in a pharmacologic screen from a panel of rationally designed peptidyl–aldehyde inhibitors using substrates specific for the chymotryptic activity of the iP [44]. IPSI-001 showed antiproliferative and apoptotic effects in MM cell lines and purified patient plasma cells. Samples from patients with diffuse large B-cell non-Hodgkin lymphoma (NHL), chronic lymphocytic leukemia (CLL), and AML were also sensitive to IPSI-001 treatment, as well as samples with acquired bortezomib resistance [44].

**PR-924** was discovered by Parlati et al. [41] in a pharmacologic screen designed to find analogs of carfilzomib that preferentially target β5i. In this initial study, a one-hour pulse treatment with PR-924 (which induced a β5i inhibition of 90%) could not induce apoptosis in MM, B-lymphoma, or T-lymphoma cell lines. In contrast, when combined with a selective inhibitor of the constitutive β5 subunit or with the genetic knockdown of β5, these cells became sensitive to PR-924. More recent reports found that PR-924 treatment showed cytotoxicity in MM cell lines and primary patient cells without affecting normal peripheral blood mononuclear cells (PBMCs) [145]. In vivo treatment with PR-924 inhibited tumor growth and prolonged survival of human MM xenograft murine models. In agreement with these data, treatment of MM, T-cell ALL (T-ALL), and AML cell lines with PR-924 led to antiproliferative and apoptotic effects [79]. The treated cells acquired drug resistance after 3 months of treatment with increasing concentrations of PR-924. The β5i/*PSMB8* subunit was checked for mutations but none were found, in contrast to β5/PSMΒ5, where the same mutations as in bortezomib-resistant cells were found. PR-924-resistant cells exhibited a 2.5-fold upregulation of cP subunits, whereas iP expression decreased 2-fold [79].

**LU-035i** is one of the most specific inhibitors for the β5i subunit; it was designed based on ONX-0914 and PR-924 through a structure-based design approach [154]. Apart from LU-035i, de Bruin et al. [154] also developed LU-015i, another potent β5i inhibitor, and LU-001i, a β1i inhibitor based on a previously designed β1 inhibitor, NC-001 [164]. A novel strategy for cancer treatment is the conjugation of cytotoxic agents to a peptide with a high affinity for tumor-specific proteins [165]. Since MM cells have a high expression of iPs, conjugating an iP inhibitor that covalently binds to the iP may represent a good strategy for the selective delivery of cytotoxic drugs that will otherwise show unspecific binding. LU-035i has been used to selectively target MM cells for treatment with doxorubicin [161]. Further support for this design is the observation that proteasome inhibition synergizes with doxorubicin therapy [166]. Treatment of a MM human cell line with the conjugate LU-035i–doxorubicin increased cell death in sensitive and carfilzomib-resistant cells [161].

**HT2210 and HT2106** are two oxathiazolones that were found to be active against iP [155]. Treatment with HT2210 or HT2106 inhibited β5i activity (90% decreased activity at 10nM) of an NHL human cell line and induced an accumulation of polyubiquitinylated proteins [155].

**1-CA and 4-CA** were developed as peptide–ketoepoxides related to the cP inhibitor carfilzomib with selectivity for the β5i subunit [156]. Using an AML human cell line, the effects on cell viability of 1-CA and 4-CA treatments were determined, and no significant effects were observed.

**KZR-616** was derived from the iP inhibitor ONX-0914 to increase its affinity not only to the β5i subunit but also to the β1i subunit to obtain a double-inhibitor [158]. In subsequent studies, it was concluded that treatment of a murine model of arthritis with single inhibitors of β5i/*LMP7* or β1i/*LMP2* was not sufficient to have an effect on disease progression, while double inhibition using KZR-616 resulted in a reduction of the disease burden [167]. Moreover, the solubility of KZR-616 is 14,000-fold higher than ONX-0914, which makes it a good candidate for clinical trials [167]. KZR-616 was the first iP inhibitor to successfully complete a phase I clinical trial for the treatment of autoimmune and inflammatory diseases [168] and enter a phase II study for the treatment of patients with inflammatory myopathies such as polymyositis or dermatomyositis and a phase Ib/II trial for the treatment of systemic lupus erythematosus and lupus nephritis [169,170]. To date, there have been no data published on the use of KZR-616 in hematopoietic malignancies, but the promising data obtained on autoimmune and inflammatory diseases suggest the possibility of further exploring its potential for the treatment of hematologic malignancies.

**M3258** was synthesized using the α-aminoboronic acid scaffold as a starting point through the optimization of potency and selectivity to the β5i subunit [160]. From a pool of DNA-barcoded MM, leukemia and lymphoma cells that were treated with M3258, a subset of cell lines, were discovered to respond to M3258 treatment. Reductions in cell viability of more than 50% [162] could be observed. Moreover, in vivo treatment in several MM xenograft models, including models resistant to bortezomib treatment, demonstrated anti-tumor activity [138,160,171]. M3258 has entered a phase I clinical trial for the treatment of MM as a single agent or in combination with dexamethasone [163].

## 5. Pathways Affected by Immunoproteasome Inhibition

Treatment with non-selective PIs has been shown to impact different pathways. First of all, proteasome inhibition confers severe proteotoxic stress to cells, affecting protein quality control in the endoplasmic reticulum (ER) and the cytosol. This is manifested by the accumulation of ubiquitin conjugates, the activation of unfolded and integrated stress responses [172], and the activation of the transcription factor Nrf1/Tcf11 [89,90]. Proteasome inhibition with bortezomib has been suggested to prevent the degradation of IkB, an inhibitor of the nuclear factor-κB (Nf-κB) pathway, blocking this pathway and, consequently, the activation of downstream pathways such as cytokine and survival factor production. However, other studies have shown that bortezomib can increase Nf-κB activation in MM cell lines and patient samples [173,174]. The pro-apoptotic protein NOXA has also been demonstrated to be an important mechanism of PI treatment. NOXA becomes upregulated after bortezomib treatment [175], inducing apoptosis by binding to the anti-apoptotic proteins of the Bcl-2 subfamily or other factors. Moreover, other effects of non-selective PIs include the induction of cell cycle arrest [176], stimulation of angiogenesis [177], and increased DNA repair [178]. As stated before, PIs inhibit both cPs and iPs, making it difficult to determine what part of their effect is due to the specific inhibition of iPs. Consequently, so far, it remains elusive which intracellular pathways are affected by specific cP or iP inhibition. Because of the wide range of substrates, the turnover of which may be accelerated by iPs compared to cPs [81], iP inhibition most likely affects multiple pathways rather than one selective target. Therefore, differential sensitivity to iP inhibition can be expected, depending on the cell type and the genetic cellular background.

Supporting the notion that the cellular processes affected by iPs may vary depending on cell type, a co-regulation of genes involved in immune processes with iP expression was found in non-MLLr AML cell lines (which are insensitive to iP inhibition), while in MLLr cell lines, iP genes were co-regulated with genes involved in cell metabolism and proliferation, mitochondrial activity, and stress response [136]. 

Effects on the phosphorylation of the three main MAPK pathways (ERK1/2, p38/SAPKs, and JNKs) have been observed following iP inhibition. Similarly, in lymphocytes derived from patients with mutations in *PSMB8*, increased levels in phosphorylation of p38 could be detected [121]. In primary human and mouse lymphocytes, iP inhibition reduced ERK phosphorylation [179], and, similarly, inhibition of *LMP7* in macrophages resulted in consistent impairment of ERK1/2 and p38 phosphorylation [180]. Likewise, bone-marrow-derived macrophages showed reduced activation of all three pathways [181]. In contrast, JNK was found to be activated after iP inhibition in MM cell lines [44]. This finding supports the notion that the consequences of iP inhibition are cell type- and oncogene-dependent.

Immunoproteasome may also influence cytokine production. In PBMCs and AML human cell lines, in vitro iP inhibition reduces the production of several cytokines, including IL-23, IL-6, IL-2, TNF-α, and INF-γ [45,156]. In line with these results, bone-marrow-derived dendritic cells from *LMP2*-deficient mice exhibited substantially reduced levels of IFN-γ, IL-6, IL-1b, and TNF-α upon infection with influenza A virus [182], and iP inhibition in autoimmune and inflammatory mouse disease models resulted in decreased production of pro-inflammatory cytokines as well [45,137,151]. Moreover, mutations of the β5i subunit led to an increase in IL-6 in the serum, skin, and B-cells of patients harboring this mutation [121,122]. Transcription factor NF-κB induces the expression of various pro-inflammatory cytokines, the activation of which, by proteasomes, has been clearly established [183]. In this regard, various studies have aimed at determining how iP inhibition or inactivation impacts this pathway. NF-κB is sequestered in the cytoplasm and is inhibited when bound to proteins of the IκB family, which are degraded by the proteasome. The protein turnover of IκB is higher in cells expressing iP subunits than in cells mainly expressing cPs [81]. Moreover, a reduction in the activation of the NF-κB pathway was observed in *LMP7*-deficient mice over the course of CVB3 infection [80] and in MM cell lines after iP inhibition [44]. However, other studies in macrophages, cardiomyocytes, and lymphocytes failed to detect an effect on NF-κB after iP inhibition [179,180]. For that reason, it remains a matter of debate whether the reduction in cytokine production, observed after iP inhibition, relies on impaired NF-κB signaling.

The observation that iP inhibition induces apoptosis in cancer cells has led to the investigation of its effects regarding the activation of the apoptotic machinery. Immunoproteasome inhibition led to an increase in apoptosis in MM cell lines through the activation of intrinsic (caspase-9-mediated) and extrinsic (caspase-8-mediated) apoptotic pathways that merged in the activation of the common effector caspase-3 [44,145]. In agreement with the activation of the intrinsic apoptotic pathway, cleavage of poly(ADP)-ribose polymerase (PARP), translocation of cleaved-BID to mitochondria, accumulation of proapoptotic Bax, and cytochrome c release were observed following iP inhibition [44,145]. In contrast, T-cells did not show increased apoptosis and were able to recover from the proteostatic stress response following iP inhibition by the activation of Nrf1 [179], a transcription factor that induces expression of the cP subunits [89]. This increase in cP expression may be able to restore—at least in part—the cell’s homeostasis.

## 6. Discussion

The development of new targeted and personalized therapeutic strategies has improved the survival of patients with hematopoietic cancers. The prevalence of hematologic malignancies has increased due to demographic change and an aging population. However, the long-term survival of older patients remains rather low [184,185]. The UPS, due to its crucial role in protein homeostasis, has been associated with tumorigenesis [31,32,33,34] and moved into focus as a putative therapeutic target for the treatment of hematologic malignancies [35,36].

Different enzymatic processes that integrate the UPS beyond proteasome inhibition can be targeted for therapeutic intervention. Examples include ubiquitin activation by E1 enzymes that can be blocked by the small molecule PYR41, which has demonstrated antileukemic activity in mouse models [186]; E1 enzyme NEDD8-activating enzyme (NAE) can be inhibited to blunt the activation of the NF-κB pathway, DNA damage, and cell death in lymphoma and AML xenograft models [187,188]. Despite the fact that several studies have linked E2s to cancer and the efforts to develop strategies that target these enzymes, there are currently no reported therapies involving E2 enzymes [189]. The high substrate specificity that characterizes E3 enzymes [3] has been exploited for the development of a novel strategy for drug discovery that allows the marking of essential proteins to cancer biology that have no targetable catalytic activity. Targeted protein degradation employs small molecules that act as “molecular glue” to recruit a specific target protein to E3 enzymes, leading to its ubiquitination and subsequent degradation by the proteasome [190]. Thalidomide and thalidomide analogs—lenalidomide and pomalidomide—are the first approved degrader drugs to target an E3 ubiquitin-ligase. Cereblon (CRBN) is the substrate adaptor of the CRL4^CRBN^ cullin-ring ligase E3 enzyme and has been identified as the target of thalidomide [191]. In MM cells, IKZF1 and IKZF3 are selectively ubiquitinylated by CRL4^CRBN^ in the presence of lenalidomide, leading to antitumor effects [192,193]. Lenalidomide is also highly effective in myelodysplastic syndrome (MDS) with the deletion of chromosome 5q, where it induces the ubiquitination of casein kinase 1A1 (CK1α) by CRL4^CRBN^ [194].

While targeting E1, E2, or E3 enzymes can provide specificity for individual substrates, targeting the proteasome inhibits the final step of protein degradation. Non-selective PIs (targeting cPs and iPs) have demonstrated efficacy against hematologic malignancies. Toxicities and therapy resistance associated with their use have prompted the development of iP inhibitors. Despite promising results observed in pre-clinical experiments, only one iP inhibitor is currently in clinical trials for the treatment of myeloma. The characteristics that make them suitable for inflammatory diseases may also be suitable for the treatment of hematologic cancers.

Selective inhibition of more than one iP subunit may increase efficacy in clinically relevant settings. For non-selective PIs, it has been shown that β5 inhibition alone is sufficient to induce cytotoxicity in PI-sensitive but not PI-resistant cells. Conversely, the combination of β1/β2 with β5 inhibition is also effective in PI-resistant cells [195]. Along these lines, selective iP inhibition and the inhibition of more than one immunosubunit may also improve inhibitory efficacy. Consistent with this suggestion, KRZ-616, a dual inhibitor of β5i and β1i subunits, was able to effectively reduce the inflammatory phenotype in autoimmune disease; in contrast, treatment with iP inhibitors of the β1i and β5i subunits alone did not have any beneficial effect [167].

Taken together, iP inhibition represents a promising therapeutic avenue for inflammatory diseases and hematopoietic cancers. 

## Figures and Tables

**Figure 1 cells-10-01577-f001:**
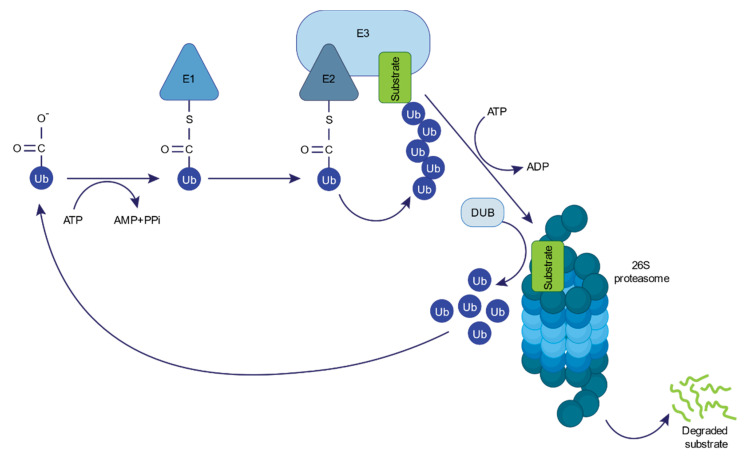
The ubiquitin–proteasome system (UPS). UPS-mediated protein degradation requires ubiquitin-activating enzymes (E1s), ubiquitin-conjugating enzymes (E2s), ubiquitin-ligase enzymes (E3s), and the 26S proteasome. Within the UPS, a reversed reaction of protein deubiquitylation catalyzed by deubiquitinating enzymes (DUBs) is also performed.

**Figure 2 cells-10-01577-f002:**
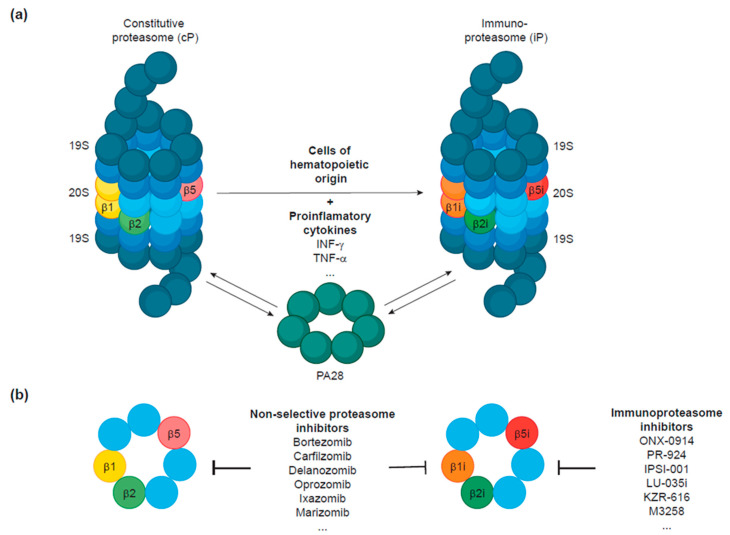
(**a**) Structure of the constitutive proteasome (cP) and the immunoproteasome (iP). Following stimulation with pro-inflammatory cytokines, the immunosubunits (β1i, β2i, and β5i) are preferentially incorporated into proteasomes to the detriment of the constitutive ones (β1, β2, β5). The iP is constitutively expressed in cells of hematopoietic origin under normal conditions. Independently of INF-γ-induced iP expression, expression of the regulatory particle PA28 is also upregulated by INF-γ and can bind to one or both ends of the 20S core particle of the cP or the iP with the same affinity. (**b**) Inhibitors of the proteasome can be designed to target both cPs and iPs or to selectively target the iP.

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
