# Peer review of "Immunoproteasome Function in Normal and Malignant Hematopoiesis"

_cells, 2021, doi:10.3390/cells10071577_

Round 1

Reviewer 1 Report

The review article entitled ‘Immunoproteasome function in benign and malignant hematopoiesis’ by Tubio-Santamaria covers the current knowledge of immunoproteasome (iP) involvement in heamatopoiesis and leukocyte biology as well as the potential of selective immunoproteasome inhibition for the therapy of leukemias. Given that normal T cells, B cells, NK cells, monocytes as well as their precursors and delineated neoplasms express high levels of immunoproteasomes, iP inhibition holds a lot of therapeutic potential which has remained underexplored. Moreover, I am not aware of reviews that have focused on this topic rendering the topic of this review timely and original. The review is very well written and well-structured making it a pleasure to read this piece. The cited articles are up to date and include classical original studies as well as latest publications on this topic. The citations are well chosen and balanced. They do not overrepresent the authors’ own original works. The ubiquitin-proteasome system is first introduced so that it is understandable to non-expert readers. Then the structure and enzymatic properties of the iP are introduced followed by its expression pattern which is pivotal for its suitability as a drug target. The brief chapter on disease causing mutations of proteasome subunits in humans is a bit out of scope but as the Krueger lab has made major contributions to this field and as these fairly recently reported diseases are rare and still not well known even to readers within the protein degradation area, I think that this chapter in its brevity is justified. A central topic-relatedness has the chapter 3 on pro- and anti-tumoral properties of the immunoproteasome. Also well done and cutting edge is chapter 4 on the currently available selective iP inhibitors which benefits from brief paragraphs on each inhibitor flanked by their listing in tables 1 and 2. Chapter 5 introduces more recent studies on potential mechanisms how iP inhibition works mechanistically emphasizing that there is still a lot to be discovered. The Discussion extends the look to other ubiquitin-proteasome related therapeutic approaches and clinical aspects and may be extended in the chapter heading to ‘Discussion and Outlook’.

In spite of all these positive aspects in design and comprehensive subject coverage, there are a few flaws which need to be corrected before it can be accepted for publication:

  • Figure 2a shows the catalytically active subunits of the 26S cP and iP in colours but the order is not corrected there are two subunits in between b2/b2i and b5/b5i, namely b2 and b4 and not only one. Moreover, the order of subunit rearrangements in the two facing b-rings is not parallel but anti-parallel with the b1/b1i subunits juxtaposed. Also in Figure 2b there needs to be a further subunit inserted between b2/b2i and b5/b Please correct this figure accordingly.
  • I am aware that the paper of the Rock group on gene targetd mice lacking b1i, b2i, and b5i claims that the phenotype of these mice is much stronger than that of e.g. b5i single knock out mice. However, when one compares the immunological phenotypes of these mice (e.g. in viral infection models but also otherwise) they are quite similar to b5i deficient mice. This is not unexpected because in b5i knock out mice also b1i and b2i are much less incorporated into iP complexes. I would therefore tune down the statement on the difference of single and triple knock out mice (line 157).
  • In lanes 166 to lane 168 it is stated that ‘The IP has been shown to prevent the accumulation of harmful protein aggregates under cytokine-induced oxidative stress due to an increased efficiency in the degradation of poly-ubiquitylated proteins’. Here it remains unclear in comparison to what the efficiency is increased – if this is cP, it should be stated. The authors cite ref. 79 in which data are presented that iP degrades ubiquitin conjugates faster than cP. This paper has been challenged by Nathan et al. (2013) Cell 152:1184-1194 and also in the recently published reference 82 where it is shown that the isolated 20S iP degrades oxidized proteins faster than 20S cP in vitro but that the kinetic of ubiquitin-dependent degradation of substrates in cells is not affected by iP expression. Moreover, a recent study by Basler et al. (2001) Immunol. 206:1697-1708 shows that in tissues of virus infected mice experiencing systemic IFNg stimulation and protein folding stress no difference in the unfolded protein reponse and ubiquitin conjugate accumulation was observed when b5-deficient and wild type mice were compared. It is probably beyond the scope of this review to go in detail about this controversial discussion but biased or unclear statement should be avoided. I suggest that the critical sentence could be changed to ‘…an increased efficiency in protein degradation.’ or it could be stated that this issue is still controversial and then the study of Nathan et al (2013) ought to be cited along with ref. 79.
  • In lane 183 I suggest that the authors’ own finding that Nrf1 induces all 20S proteasome subunits but none of the ‘immuno-subunits’ b1i, b2i, b This is a striking finding confirmed by the Deshaies lab and worth pointing out.
  • Typos: lane 254 ‘enhanced’, lane 325 ‘were’
  • At last my suggestion for a change in title from ‘Immunoproteasome function in benign and malignant hematopoiesis’ to ‘Immunoproteasome function in normal and malignant hematopoiesis’. My understanding is that the term ‘benign’ is used to distinguish less dangerous tumors from malignant ones. If I understand the authors right, they want to cover normal and neoplastic hematopoiesis.

Author Response

We would like to thank the referees for their valuable comments and suggestions. Please, find below a response to each.

  • Reviewer #1: Figure 2a shows the catalytically active subunits of the 26S cP and iP in colours but the order is not corrected there are two subunits in between b2/b2i and b5/b5i, namely b2 and b4 and not only one. Moreover, the order of subunit rearrangements in the two facing b-rings is not parallel but anti-parallel with the b1/b1i subunits juxtaposed. Also, in Figure 2b there needs to be a further subunit inserted between b2/b2i and b5/b Please correct this figure accordingly.

We apologize for the mistake and thank the reviewer for the comment. We updated figure 2 accordingly (see Fig.2 in the annotated manuscript, page 4).

  • Reviewer #1: I am aware that the paper of the Rock group on gene targeted mice lacking b1i, b2i, and b5i claims that the phenotype of these mice is much stronger than that of e.g. b5i single knock out mice. However, when one compares the immunological phenotypes of these mice (e.g. in viral infection models but also otherwise) they are quite similar to b5i deficient mice. This is not unexpected because in b5i knock out mice also b1i and b2i are much less incorporated into iP complexes. I would therefore tune down the statement on the difference of single and triple knock out mice (line 157).

We have clarified the part focusing on the different phenotypes for the single and the triple deficient iP mice (see annotated manuscript, lines 160-163).

  • Reviewer #1: In lanes 166 to lane 168 it is stated that ‘The IP has been shown to prevent the accumulation of harmful protein aggregates under cytokine-induced oxidative stress due to an increased efficiency in the degradation of poly-ubiquitylated proteins’. Here it remains unclear in comparison to what the efficiency is increased – if this is cP, it should be stated. The authors cite ref. 79 in which data are presented that iP degrades ubiquitin conjugates faster than cP. This paper has been challenged by Nathan et al. (2013) Cell 152:1184-1194 and also in the recently published reference 82 where it is shown that the isolated 20S iP degrades oxidized proteins faster than 20S cP in vitro but that the kinetic of ubiquitin-dependent degradation of substrates in cells is not affected by iP expression. Moreover, a recent study by Basler et al. (2001) 206:1697-1708 shows that in tissues of virus infected mice experiencing systemic IFNg stimulation and protein folding stress no difference in the unfolded protein reponse and ubiquitin conjugate accumulation was observed when b5-deficient and wild type mice were compared. It is probably beyond the scope of this review to go in detail about this controversial discussion but biased or unclear statement should be avoided. I suggest that the critical sentence could be changed to ‘…an increased efficiency in protein degradation.’ or it could be stated that this issue is still controversial and then the study of Nathan et al (2013) ought to be cited along with ref. 79.

We thank the reviewer for this valuable comment. We have included the reference Nathan et al. (2013) Cell 152:1184-1194 and modified the sentence in agreement with the statement provided by reviewer #1 (see annotated manuscript, lines 172-182).

  • Reviewer #1: In lane 183 I suggest that the authors’ own finding that Nrf1 induces all 20S proteasome subunits but none of the ‘immuno-subunits’ b1i, b2i, b This is a striking finding confirmed by the Deshaies lab and worth pointing out.

We thank the reviewer for detecting this weakness.  We have added 2 relevant references (PMID:20932482, PMID:12676932) addressing this point (see annotated manuscript, line 190-193).

  • Reviewer #1: Typos: lane 254 ‘enhanced’, lane 325 ‘were’

The typo “enhanced” was corrected (see annotated manuscript, line 266); for line 325 we chose “was” over “were”, since the verb refers to only one cell line (line 351).

  • Reviewer #1: At last my suggestion for a change in title from ‘Immunoproteasome function in benign and malignant hematopoiesis’ to ‘Immunoproteasome function in normal and malignant hematopoiesis’. My understanding is that the term ‘benign’ is used to distinguish less dangerous tumors from malignant ones. If I understand the authors right, they want to cover normal and neoplastic hematopoiesis.
  • We thank the reviewer for this comment. From the view of a clinical hematologist and oncologist, the word benign is not restricted to behavior of tumor cells, but rather distinguishes malignant from non-malignant cells.

    However, we agree with the reviewer that this aspect may be misleading to the readership and adapted the title accordingly to: ‘Immunoproteasome function in normal and malignant hematopoiesis’.

Reviewer 2 Report

Authors review data on immunoproteasome (iP) function and inhibition in healthy hematopoietic system and cancers. Unfortunately, a limited description of normal cells iP function and targeting is reported with major data on iP role and targeting on malignant cells. It is opinion of this referee that a major relevance on iP function and its targeting on normal cells should be included otherwise I suggest to modify title including cancer cells as well. Moreover, comprehensive list of most effective compounds on normal cells should be also described with a new table.

In general, this is an easy to ready and up-dated review on iP role and its targeting on hematologic cells which deserves fully consideration for its publication.

Author Response

We would like to thank the referees for their valuable comments and suggestions. Please, find below a response to each.

  • Reviewer #2: Unfortunately, a limited description of normal cells iP function and targeting is reported with major data on iP role and targeting on malignant cells. It is opinion of this referee that a major relevance on iP function and its targeting on normal cells should be included otherwise I suggest to modify the title including cancer cells as well.

We thank the reviewer for the comments. We have included one more paragraph about iP inhibition in normal hematopoietic cells (see annotated manuscript, lines 314-323).

In the revised version, the function of iP in normal cells has now been addressed to the best of our knowledge (explanation of the effects of iP deficiency in mouse models, lines 160-163; non-immune functions of the iP in different hematopoietic cell types including T cells and macrophages , lines 164-169; expression of iP in cells of hematologic origin, lines 196-200; inhibition of iP in normal hematopoietic cells, lines 314-323); pathways affected by iP inhibition in different cell types including lymphocytes, macrophages and dendritic cells (lines 467-469 and 474-480).

  • Reviewer #2: Moreover, comprehensive list of most effective compounds on normal cells should be also described with a new table.

We agree with the reviewer that a list of effective compounds on normal cells would increase the value of our review. However, available experimental data on iP inhibition in normal cells are limited. Usually, proteasome inhibitors are used in models of disease and cancer. As stated in comment 2.1., we have included one paragraph with additional information on iP inhibition in normal hematopoietic cells (see annotated manuscript, lines 314-323).

Reviewer 3 Report

This is a comprehensive review focused on the immunoproteasome. The review provides a succinct intro on ubiquitin-proteasomes system and explains difference between immune and constitutive proteasomes. It then goes into detail on roles of the immune proteasome in general as well as related to cancer and inflammatory disease. Next, a series of recent drugs and compounds in development are highlighted and discussed, this list is more a summary than providing big insights, but it is helpful to have the comprehensive list.

Overall, these parts are written well and presented nicely and the author shave good knowledge of the literature. However, I was confused on some key part related to the general CP inhibitors vs iCP specific inhibitors. If I am correct the authors provide two separate rationales for specific iCP inhibitors within the review (which are not mutually exclusive), but I feel they are somewhat mixed up in the presentation.

Argument 1. iCP is strongly expressed in hematopoietic cells (and some other cell types), but not in many other cell types (see line 295/296). Therefore, specifically inhibiting iCP would be more specific as it will not inhibit proteasomes in all other cell types and thus reduce changes of side-effects. In this argument additional specific functions of iCP that are inhibited might be less relevant.

Argument 2. Targeting iCP instead of cCP + iCP is more specific and one might reduce side effects within one cell type by not inhibiting all proteasome functions, but specific ones (see e.g. line 287). At several places the authors argue that iCP inhibition would be less non-selective, e.g. “Inhibiting cCP and iCP might lead to non-selective inhibition of degradations, targeting iCP more selective”. Here distinguishing between cCP and iCP functions is more relevant.

In either case it would be relevant to discuss the models that currently exists of how PI is effective in MM treatment, because are those inhibited functions conducted by iCP, cCP, or both? From the review it is unclear to me if PIs work because they target a cCP function or is the cCP inhibition irrelevant and is all effectiveness in MM treatment based on PIs ability to inhibit iCP? On one hand the authors suggest the later as they discuss specific roles of iCP in section 5 and do not discuss general roles of cCP with regards to treatment and drug effectiveness. Is my understanding of an important contribution of ERAD inhibition in the effectiveness of PI against MM outdated? Or does the expression of iCP subunits make most CP in hematopoietic iCP and inhibiting iCP would thus also inhibit ERAD? If that is true, is the benefit/effectiveness of iCP than mainly based on argument 1 and less so on specific functions iCP might have?  

Maybe in the end it turns out to be a combination of both. Either way, if my understanding of the ideas and logic is correct it would help to discuss this more explicit and also to discuss how iCP and cCP differ or conduct the same function in MM and how inhibition of either would impact cells. It seems impossible to discuss iCP inhibition in MM without discussing the ideas of how PI kill MM cells in the first place and I am missing that discussion.

I think including the perspective of the authors on these aspects more clearly would strengthen the review.

Some small specific comments:

Line 40 “allowing for” rephrase…. The characterization is not what allows for the selectivity

Line 43 If you mention M1, it might be helpful to note to readers that ubiquitin is attached to proteins using the C-terminal Glycine of Ubiquitin.

Line 51: define constitutive proteasome and also provide some information on evolution or define clearly that human is discussed.

Line 54: include information that these three are the only subunits with proteolytic activity.

Line 56 delete “or” ?

Line 65 replace “this” with PA200 to avoid confusion with alpha4.

Lin 68 some recent PA200 structure papers raise some question about the acetylated binding site and this premise?

Sentences 73-75 please rephrase not intrinsically logic

Line 77. While authors use “may” the sentences is suggesting that it has been demonstrated that UPS role in hematopoietic malignancies formation from the HSC. However, if I am correct references are more general related to UPS – cell cycle. Add specific references or I would suggest rephrasing to be more clear.

Line 97 malignant transformation is an early event in cancer. So, targeting that seems useless as it would require treatment prior to cancer development? Do PI work because they target this?

These two points go to a bigger issue I was unclear about, the author seems to focus on UPS in the development of the cancer. Preventing formation might be hard, unless everybody would be pretreated. Once a cancer formed, why do proteasome inhibitors work in treatment?

Line 92 why use “may” here? The reference shows that, so if the authors wonder about the relevance of those data in vivo provide some context to what the limitations of that study were.

Line 110-115 Included information on Rpn1 as ubiquitin receptor seems prudent.

Line 123 incorporation and line 125 replace is with regard to newly formed proteasomes, might help to clarify even though the POMP sentence that follows indicates that.

Line 151: efficiency of peptide formation or efficiency of formation of peptides (more) suitable for MHC I loading?

Line 170 reference to a paper from 2001 is not suitable to support “current debate”

The section 5 talks about immunoproteasome pathways comes right after specific iCP inhibitors, but earlier there was a focus on bortezomib and the likes vs more selective iCP inhibitors. Wouldn’t the processes described here be equally impacted by either drug and is it truly relevant that iCP drugs target iCP specific processes, or does the iCP in MM e.g. also play a role in ERAD and the only true relevant aspect is that CP is inhibited? Line 428 bortezomib and other CP inhibitors presumably would induce the same effect since they also inhibit iCP, so how does this make iCP vs cCP+iCP inhibitors better? This section #5 that talks about immunoproteasome pathways comes right after specific iCP inhibitors, with the overall aim and focus on drugs this section should then also discuss general PIs.  

Author Response

We would like to thank the referees for their valuable comments and suggestions. Please, find below a response to each.

Overall, these parts are written well and presented nicely and the author shave good knowledge of the literature. However, I was confused on some key part related to the general CP inhibitors vs iCP specific inhibitors. If I am correct the authors provide two separate rationales for specific iCP inhibitors within the review (which are not mutually exclusive), but I feel they are somewhat mixed up in the presentation.

Argument 1. iCP is strongly expressed in hematopoietic cells (and some other cell types), but not in many other cell types (see line 295/296). Therefore, specifically inhibiting iCP would be more specific as it will not inhibit proteasomes in all other cell types and thus reduce changes of side-effects. In this argument additional specific functions of iCP that are inhibited might be less relevant.

Argument 2. Targeting iCP instead of cCP + iCP is more specific and one might reduce side effects within one cell type by not inhibiting all proteasome functions, but specific ones (see e.g. line 287). At several places the authors argue that iCP inhibition would be less non-selective, e.g. “Inhibiting cCP and iCP might lead to non-selective inhibition of degradations, targeting iCP more selective”. Here distinguishing between cCP and iCP functions is more relevant.

In either case it would be relevant to discuss the models that currently exists of how PI is effective in MM treatment, because are those inhibited functions conducted by iCP, cCP, or both? From the review it is unclear to me if PIs work because they target a cCP function or is the cCP inhibition irrelevant and is all effectiveness in MM treatment based on PIs ability to inhibit iCP? On one hand the authors suggest the later as they discuss specific roles of iCP in section 5 and do not discuss general roles of cCP with regards to treatment and drug effectiveness. Is my understanding of an important contribution of ERAD inhibition in the effectiveness of PI against MM outdated? Or does the expression of iCP subunits make most CP in hematopoietic iCP and inhibiting iCP would thus also inhibit ERAD? If that is true, is the benefit/effectiveness of iCP than mainly based on argument 1 and less so on specific functions iCP might have?

This is an interesting view on PI treatment. There are a couple of arguments, why PI treatment works quite well in MM. 1. Basically, PI treatment targets all proteasome types in all cells and all functions of the proteasome. For systemic Bortezomib (BTZ) application by infusion in MM patients, it is clear that blood cells are targeted first with a high efficacy due to pharmacokinetics/ dynamics and thus with low side effects on other tissues. The blood brain barrier almost prevents targeting of cells of the CNS. For other tissues, the BTZ concentration is low with low side effects by dilution (blood cells take everything).2. Leukocytes produce a lot of protein/ peptide factors, which take the secretory pathway (cytokines, surface markers, receptors, antibodies …) and thus require an efficient ER-protein quality control including ERAD for their function. Proteasomal degradation is the last part of ERAD and a sensitive chain link for targeting by PIs here. From our own preliminary studies we know that immunoproteasome inhibition of leukocytes and leukemia cells induces the unfolded protein response and ERAD as well confirming that iP inhibition represents a good option. 3. Leukocytes including MM cells mainly express immunoproteasomes. BTZ-resistant MM cells show manipulation of proteasome composition and expression. Molecular mechanisms of BTZ resistance are not fully understood; however, three major features are observed in patients with relapses, namely, (i) genetic alterations in standard β5, (ii) altered proteasome subunit compositions, and (iii) induction of proteasome subunits. Genetic analysis of BTZ-resistant cells revealed a hot spot for mutations in exon 2 of PSMB5 encoding the β5 subunit. All mutations affect the BTZ binding site of β5, alanine in position 49, directly or in close proximity to, and prevent efficient binding of BTZ clearly recommending the use of iCP inhibitors. We included a paragraph discussing this (see annotated manuscript, lines 299-305). 4. The combination of PI with inhibitors of other parts of the UPS such as inhibitors of the E3 ubiquitin ligase cereblon are discussed in lines 525-532 of our manuscript.

  • Reviewer #3: Maybe in the end it turns out to be a combination of both. Either way, if my understanding of the ideas and logic is correct it would help to discuss this more explicit and also to discuss how iCP and cCP differ or conduct the same function in MM and how inhibition of either would impact cells. It seems impossible to discuss iCP inhibition in MM without discussing the ideas of how PI kill MM cells in the first place and I am missing that discussion.

I think including the perspective of the authors on these aspects more clearly would strengthen the review.

We agree with the reviewer that a discussion on the effects of non-selective PIs is missing. We have included one paragraph inside section 5 explaining the effects of PIs (see annotated manuscript, lines 435-449).

Some small specific comments:

  • Reviewer #3: Line 40 “allowing for” rephrase…. The characterization is not what allows for the selectivity

The sentence was re-phrased accordingly (see annotated manuscript, line 40).

  • Reviewer #3: Line 43 If you mention M1, it might be helpful to note to readers that ubiquitin is attached to proteins using the C-terminal Glycine of Ubiquitin.

We thank the reviewer for the comment. Information about how ubiquitin binds to proteins was included (see annotated manuscript, line 41).

  • Line 51: define constitutive proteasome and also provide some information on evolution or define clearly that human is discussed.

We thank the reviewer for this comment. A more exhaustive explanation about the proteasome structure is described in the section 2.1. Information on the evolution of the different proteasome variants has been added (see annotated manuscript, lines 57-59 and 65-66).

  • Reviewer #3: Line 54: include information that these three are the only subunits with proteolytic activity.

We agree with reviewer #3 that this piece of information needs to be included. In the revised version of the manuscript, we have clarified the main catalytic subunits (see annotated manuscript, line 55-56).

  • Reviewer #3: Line 56 delete “or”?

We apologize for the mistake. The typo was corrected (see annotated manuscript, line 59).

  • Line 65 replace “this” with PA200 to avoid confusion with alpha4.

We thank reviewer #3 for indicating this aspect. We have re-phrased and clarified this aspect in the revised version of our manuscript (see annotated manuscript, line 69).

  • Line 68 some recent PA200 structure papers raise some question about the acetylated binding site and this premise?

We included a more recent reference (PMID: 30104204) that confirms the role of PA200 in promoting degradation of acetylated core histones (see annotated manuscript, line 73).

  • Sentences 73-75 please rephrase not intrinsically logic

We thank reviewer #3 for the comment. The sentence was changed in order to make it clearer (see annotated manuscript, lines 78-81).

  • Line 77. While authors use “may” the sentences is suggesting that it has been demonstrated that UPS role in hematopoietic malignancies formation from the HSC. However, if I am correct references are more general related to UPS – cell cycle. Add specific references or I would suggest rephrasing to be more clear.

The sentence was changed accordingly to be more assertive and more general in regard to the role of UPS in malignant transformation (see annotated manuscript, lines 81-84).

  • Line 77 malignant transformation is an early event in cancer. So, targeting that seems useless as it would require treatment prior to cancer development? Do PI work because they target this?

These two points go to a bigger issue I was unclear about, the author seems to focus on UPS in the development of the cancer. Preventing formation might be hard, unless everybody would be pretreated. Once a cancer formed, why do proteasome inhibitors work in treatment?

This is an important aspect raised by reviewer #3. First of all, the UPS controls proliferation of any cell by timely degradation of cell cycle regulators (i.e. cyclins, CDKs, CDK-inhibitors, p53 and other tumor suppressors, downstream transcription factors including Myc). PI treatment usually leads to cell cycle arrest at G1-S transition. Thus, all highly proliferating cells including tumor cells are targeted by PIs similar to other chemotherapeutic agents. Manipulation of proliferation control is an early event in malignant transformation and PI are potentially capable to prevent this. Moreover, the term “malignant transformation” is not only used for cancer formation from a normal cell to a tumor cell. Also, cells harboring mutations can undergo further transformation into a more malignant or aggressive version of the respective neoplasm. Likewise, cells can undergo step-wise transformation, which is described here as cancer development. The aim to prevent further disease progression from an already transformed (malignant) cell is a realistic and achievable translational goal. With regard to UPS function, several molecules of the UPS have been described as relevant for cancer initiation and progression in addition to cell cycle control mentioned above. Therefore, “prevention” may also be seen in the context of disease progression and evolution.

  • Line 92 why use “may” here? The reference shows that, so if the authors wonder about the relevance of those data in vivo provide some context to what the limitations of that study were.

The sentence was changed accordingly (see annotated manuscript, line 97-99).

  • Line 110-115 Included information on Rpn1 as ubiquitin receptor seems prudent.

We thank Reviewer #3 and apologize for the mistake. Information and the correspondent reference were incorporated in the text (see annotated manuscript, line 121).

  • Line 123 incorporation and line 125 replace is with regard to newly formed proteasomes, might help to clarify even though the POMP sentence that follows indicates that.

The sentence was re-phrased to better explain the terms incorporation and replacement (see annotated manuscript, line 127-129).

  • Line 151: efficiency of peptide formation or efficiency of formation of peptides (more) suitable for MHC I loading?

The suggested changes were incorporated (see annotated manuscript, lines 155-156).

  • Line 170 reference to a paper from 2001 is not suitable to support “current debate”

The revised version of our manuscript was adapted according to the comment of reviewer #1 (see comment 1.3) and a new reference was added (Nathan et al. (2013) Cell 152:1184-1194; see annotated manuscript, lines 172-176).

  • The section 5 talks about immunoproteasome pathways comes right after specific iCP inhibitors, but earlier there was a focus on bortezomib and the likes vs more selective iCP inhibitors. Wouldn’t the processes described here be equally impacted by either drug and is it truly relevant that iCP drugs target iCP specific processes, or does the iCP in MM e.g. also play a role in ERAD and the only true relevant aspect is that CP is inhibited? Line 428 bortezomib and other CP inhibitors presumably would induce the same effect since they also inhibit iCP, so how does this make iCP vs cCP+iCP inhibitors better? This section #5 that talks about immunoproteasome pathways comes right after specific iCP inhibitors, with the overall aim and focus on drugs this section should then also discuss general PIs.

We would like to thank reviewer #3 for this interesting discussion.

An additional sentence was included, highlighting that for non-selective PIs (i.e. Bortezomib) targeting both cP and iP, distinguishing whether their effects can be attributed to the inhibition of the cP or the iP is rather difficult (see annotated manuscript, lines 450-452).

One could hypothesize that, since hematopoietic cancer cells show high abundance of iP subunits, at least part of the effects observed when treated with non-selective PIs might be due to the inhibition of iP, however, the extent cannot be exactly quantified. Moreover, iP inhibition can be partially compensated by an increase in cP formation (see annotated manuscript, lines 501-503).

Regarding the question of reviewer #3 about the potential benefits of selective iP inhibition over combined (cP and iP) inhibition, the significant toxicity of unselective proteasome inhibitors is one major aspect. As explained in section 4 (see annotated manuscript, lines 295- 299 and 311-314) one benefit of selective iP inhibition is the fact that the iP is not constitutively express in other tissues apart from the hematopoietic system which can efficiently reduce toxicities and side effects.

We believe that the addition of information on the effects of non-selective PIs as requested by reviewer #3, may lead to better understanding of cellular pathways affected by PIs (see annotated manuscript, lines 435-449).